# Three-Dimensional Bioprinting in Cardiovascular Disease: Current Status and Future Directions

**DOI:** 10.3390/biom13081180

**Published:** 2023-07-28

**Authors:** Zhonghua Sun, Jack Zhao, Emily Leung, Maria Flandes-Iparraguirre, Michael Vernon, Jenna Silberstein, Elena M. De-Juan-Pardo, Shirley Jansen

**Affiliations:** 1Discipline of Medical Radiation Science, Curtin Medical School, Curtin University, Perth, WA 6102, Australia; jenna.beinart@student.curtin.edu.au; 2Curtin Health Innovation Research Institute (CHIRI), Curtin University, Perth, WA 6102, Australia; 3School of Medicine, Faculty of Health Sciences, The University of Western Australia, Perth, WA 6009, Australia; jack.zhao@health.wa.gov.au (J.Z.); emily.leung@health.wa.gov.au (E.L.); 4Regenerative Medicine Program, Cima Universidad de Navarra, 31008 Pamplona, Spain; mflandes@alumni.unav.es; 5T3mPLATE, Harry Perkins Institute of Medical Research, QEII Medical Centre and UWA Centre for Medical Research, The University of Western Australia, Perth, WA 6009, Australia; michael.vernon@research.uwa.edu.au (M.V.); elena.juanpardo@uwa.edu.au (E.M.D.-J.-P.); 6School of Engineering, The University of Western Australia, Perth, WA 6009, Australia; 7Vascular Engineering Laboratory, Harry Perkins Institute of Medical Research, QEII Medical Centre and UWA Centre for Medical Research, The University of Western Australia, Perth, WA 6009, Australia; 8Curtin Medical School, Curtin University, Perth, WA 6102, Australia; shirley.jansen@health.wa.gov.au; 9Department of Vascular and Endovascular Surgery, Sir Charles Gairdner Hospital, Perth, WA 6009, Australia; 10Heart and Vascular Research Institute, Harry Perkins Medical Research Institute, Perth, WA 6009, Australia; 11School of Medicine, The University of Western Australia, Perth, WA 6009, Australia

**Keywords:** 3D bioprinting, cardiovascular disease, cells, tissues, 3D printing

## Abstract

Three-dimensional (3D) printing plays an important role in cardiovascular disease through the use of personalised models that replicate the normal anatomy and its pathology with high accuracy and reliability. While 3D printed heart and vascular models have been shown to improve medical education, preoperative planning and simulation of cardiac procedures, as well as to enhance communication with patients, 3D bioprinting represents a potential advancement of 3D printing technology by allowing the printing of cellular or biological components, functional tissues and organs that can be used in a variety of applications in cardiovascular disease. Recent advances in bioprinting technology have shown the ability to support vascularisation of large-scale constructs with enhanced biocompatibility and structural stability, thus creating opportunities to replace damaged tissues or organs. In this review, we provide an overview of the use of 3D bioprinting in cardiovascular disease with a focus on technologies and applications in cardiac tissues, vascular constructs and grafts, heart valves and myocardium. Limitations and future research directions are highlighted.

## 1. Introduction

Cardiovascular disease (CVD) is the leading cause of death and is a major contributor to disability worldwide, with a prevalence of 523 million people who suffer from CVD and an estimated 18.6 million deaths annually [1]. Clinical diagnosis and management of CVD largely relies on less invasive imaging modalities, such as the use of computed tomography (CT), magnetic resonance imaging and echocardiography [2,3]. Despite being frequently used in daily practice, it is challenging to translate two-dimensional images of cardiovascular anatomy and pathology to three-dimensional (3D) structures based on these imaging modalities and this is especially obvious when visualising complex cardiovascular structures and pathologies, such as congenital heart disease, which usually involves multiple cardiac structures with a spectrum of abnormalities. Three-dimensional (3D) printing technology has overcome these limitations by providing realistic models, thus offering superior advantages over current image visualisations [4,5,6,7,8,9,10].

The use of 3D printing in cardiovascular disease has been increasingly reported in the literature, with applications mainly focusing on the following areas: education value for medical students and health professionals, preoperative planning and simulation of complex cardiovascular procedures, development of medical devices for improvement of outcomes, optimisation of CT protocols using 3D phantoms to minimise radiation exposure to patients during routine CT scans for diagnosis and follow up, and the bioprinting of scaffolds or cardiovascular constructs [11,12,13,14,15,16,17,18]. Figure 1 summarises these current applications. Three-dimensionally printed models serve as valuable tools for medical education by enhancing understanding of anatomy and pathology when compared with current education approaches. Nearly 50% of the current applications of 3D printing lie in the pre-surgical planning of cardiovascular procedures, with results from single and multi-centre studies showing changes in surgical decision-making when 3D printed models are employed [7,14,19,20,21,22,23,24,25,26,27]. Three-dimensionally printed models are also used to simulate interventional cardiology or radiology procedures, such as endovascular aneurysm repair, increasing confidence through training and practice [28,29,30,31,32,33,34,35,36,37,38,39,40].

While there is sufficient evidence available to prove that 3D printed personalised heart and vascular models play an important role in clinical applications, 3D bioprinting represents a promising field that could revolutionise the future of cardiovascular disease treatment. Three-dimensional bioprinting uses mostly the same additive manufacturing technologies but prints functional living structures from biological components such as living cells, biomaterials and growth factors [41,42]. This can eliminate the gap between the current animal study and human trials and may in future contribute to the repair or replacement of damaged tissues or organs and the optimisation of drug delivery techniques [43,44,45,46,47]. Although still in an early stage with many challenges to overcome, substantial progress has been made over the last decade in the printing of cardiovascular constructs as well as in cardiovascular regeneration [48,49,50].

This review provides an overview of 3D bioprinting in cardiovascular disease, with a focus on the applications in cardiac tissues, vascular constructs and grafts, myocardium and heart valves. We first review different bioprinting techniques, with advantages and limitations highlighted, then provide a detailed summary of 3D bioprinting and its applications in cardiac tissues, tissue-engineered vascular constructs and grafts in treating coronary artery disease and other vascular diseases. We also provide a summary of 3D printed heart valves in treating valvular heart disease, and 3D printed cardiac patch and heart model in managing myocardial infarction and heart failure. Lastly, future directions and perspectives of 3D bioprinting in cardiovascular disease are highlighted.

## 2. Three-Dimensional Bioprinting: Where Are We Now?

Heart transplant remains the gold standard treatment for selected people with end-stage heart failure [48]. However, the number of people requiring a heart transplant far outweighs the available donor hearts and the long-term outcome of heart transplantation remains uncertain, with acute rejection, cardiac allograft vasculopathy, malignancies, infection and the development of chronic kidney disease still significant complications [49,50].

Apart from organ transplantation, current treatment options for cardiac disease include cell therapy, implanted devices (stents or stent grafts), and bypass grafting [51,52,53]. Of these treatments, cell therapy using 3D bioprinting with patient-specific cells to repair damaged cardiac tissues or to replace end-stage organ failure through tissue engineering is promising [48,54]. It is also theoretically able to bypass the limitations of poor biocompatibility, biofunctionality, and immune rejection as well as the drastic shortage of organ donors [55,56,57,58].

Although recent advances in bioprinting methodologies and technologies have addressed limitations associated with bioprinting, it is still not clinically feasible to directly translate 3D printed cardiovascular tissues to patient therapy. The main limitation is the inability to construct thick tissues. Currently, only 1 mm tissues can be produced with the incorporation of multiple cell types and induction of tissue differentiation of growth factors [59,60]. A further limitation is a lower density of cells compared with native tissues, restricting the clinical value in cases such as myocardial ischemia [61].

## 3. Three-Dimensional Bioprinting—Bioinks

Three-Dimensional bioprinting involves the deposition of living cells, bioactive molecules and biomaterials in a layered pattern to allow the generation of 3D structures [62,63]. Different from the standard 3D printing process which use various printing materials from rigid to soft and elastic depending on usefulness or application, bioinks serve as a medium to deliver living cells [64]. Bioinks possess viscoelastic properties with higher water content to protect the cells during the printing procedure from external risk factors. Bioinks used for cardiac tissue engineering mainly refer to the use of hydrogels which mimic the 3D extracellular matrix, with its biocompatibility, biodegradability and mechanical support [65].

## 4. Three-Dimensional Bioprinting Technologies

Bioprinting methods include inkjet-based, extrusion-based, and light-based printing to create geometrically complex and scalable tissues.

### 4.1. Inkjet-Based Bioprinting

Inkjet bioprinters store bioink in a cartridge and a printer head, with the printer heads deformed by thermal, piezoelectric, electrostatic or electrohydrodynamic actuators to eject droplets in a controlled fashion [63]. This is a low-cost method with a high printing speed [64] and around 80–90% for fibroblast cell viability [65,66], but is limited by a low range of printable viscosities (<30 mPa.s) [67].

### 4.2. Extrusion-Based Bioprinting

Extrusion-based bioprinting is the most common technique used. Bioink is loaded into a syringe, then printed as continuous strands using mechanical or pneumatic forces [68]. This can print biomaterials with a wide range of viscosities (up to >6 × 10^7^ mPa/s) [69] and a high level of structural integrity [70]. However, cell viability decreases with increasing pressure and/or decreasing nozzle diameter [71]. Although extrusion-based bioprinting is a convenient and inexpensive platform with reasonable costs, it has limited resolution which is inferior to other bioprinting techniques.

### 4.3. Light-Based Bioprinting

Laser-assisted bioprinting projects a laser beam onto a thin layer of bioink causing a precise droplet to fall onto the printing surface [72]. Xiong et al. [73] have successfully printed both straight and Y-shaped vascular constructs; however, fibroblast cell viability was 68.1% and 70.8% for these constructs, respectively. The main advantages of laser-based bioprinting include high resolution, the ability to produce cardiac constructs with a high cell density (up to 10^8^ cells per mL), and bioprinting with low viscosity bioinks. The main disadvantages are complexity in controlling laser pulses, challenges in fabricating cell-embedding hydrogel constructs before printing and high cost of the laser system. Table 1 summarises these 3D bioprinting techniques with advantages and limitations highlighted [55,74,75,76,77,78].

## 5. Three-Dimensional Bioprinting of Cardiac Tissues

### 5.1. Human Pluripotent Stem Cells and Cardiac Tissue Engineering

Recent advancements in bioprinting of the cardiovascular system includes areas involving human pluripotent stem cells (hPSCs) and cardiac tissue engineering [79]. Promising benefits have been shown in the use of current hPSCs and cardiac tissue engineering technologies to construct biocompatible materials for the repair and regeneration of diseased cardiac tissue. Traditionally, cardiomyocytes are among those that are known to be difficult to expand and culture. As reprogramming of differentiated human cells back into a pluripotent state has been made possible in recent decades, the potential for creating patient and disease-specific stem cells in nearly any patient is becoming a possibility for the future [80,81]. Additionally, hPSC technologies show great potential over conventional 2D monolayer culture systems for studying the development of the human heart, the heart’s response to therapeutic interventions in drug testing, and for disease modelling of acquired and inherited diseases [79,82,83]. Figure 2 provides a schematic view of the application of cardiac tissue engineering for disease modelling and drug screening in genetic heart diseases. Cardiovascular tissue engineering is a rapidly evolving field of medicine which involves multiple sophisticated components such as different cell types, biocompatible scaffold materials, cell differentiation factors and growth factors [84]. As this is an area of research that is still in its infancy, there are several issues which are yet to be solved.

### 5.2. Cellular Maturity

hPSCs such as human embryonic stem cells (hESCs) and human-induced pluripotent stem cells (hiPSCs) are promising approaches for heart regeneration and have the potential to be differentiated efficiently into cardiomyocytes, vascular endothelial cells, and vascular smooth muscle cells [79]. Although there is much international research in this space, there are concerns over the inadequate maturation of the tissues created [80]. The maturity of current hPSC-derived cardiomyocytes resembles the characteristics of foetal cardiomyocytes [85].

To promote maturation, many approaches have been proposed, including the use of biochemical stimuli (such as hormonal treatments), mechanical strain stimulation, electrical stimulation, topographic cues such as patterned surfaces and substrate stiffness, and interactions with other cells or extracellular matrixes [82,83,86]. However, the optimal state of cellular maturity for hPSC cardiomyocytes is yet to be determined; immature cardiomyocytes may cause ventricular arrhythmias and other electrical disturbance, but have increased survival compared with adult cardiomyocytes upon transplantation into a host myocardium [79].

### 5.3. Microvasculature Constructs

Creating functional microvasculature constructs is a challenging area in the 3D bioprinting of complex tissues, such as those of the heart which is highly vascularised to allow for adequate gas exchange, nutrient diffusion and waste disposal [87]. As diffusion of oxygen through tissues is limited to approximately 100–200 µm, careful incorporation of microvasculature into bioprinted materials is a fundamental step for tissue survival [87]. Characteristics which are fundamental for proper microvascular perfusion to tissues include the presence of a hollow and endothelialised lumen, a hierarchy-based branched vascular network, and a complex signalling milieu [88]. Many biofabrication technologies, such as extrusion-based 3D bioprinting, induced sprouting, electrospinning, and lithographic approaches, have been explored to address these issues, each with their own advantages and disadvantages [88,89]. These techniques generally involve microfluidics-based moulding with numerous other strategies such as the addition of proangiogenic factors (e.g., vascular endothelial growth factor (VEGF)) within the biomaterial scaffolding [90]. Figure 3 is a schematic framework proposed by Seymour et al. to address the current issues in biofabricating microvasculature [88].

Despite rapid improvement in mimicking the vascular networks of the human cardiovascular system, there is yet to be developed an approach which is able to address all of the required characteristics of microvasculature in order to maintain perfusion and survival of tissues. Specifically, the spatiotemporal precision required for appropriate physiological function is still lacking [90]. Further exploration of biomaterials and bioprinting strategies to address the structural and physiological complexity of microvascular networks will be a key stepping-stone in the research field of 3D bioprinting of cardiovascular tissues.

### 5.4. Other Issues

Other issues that are faced in the 3D bioprinting of cardiovascular tissue include but are not limited to cellular heterogeneity in different anatomical regions of the heart [79], proliferation of primary cells in vitro entering senescence [86], foreign body reaction [91], and the risk of tumours such as teratomas in undifferentiated cell populations [92]. Optimization of the ratio for co-culturing different types of other cardiac cells along with cardiomyocytes is also an issue that needs to be addressed in 3D printed cardiac patches. Anil Kumar et al. have tested the feasibility of the heterocellular coupling of cardiomyocytes and fibroblasts/endothelial cells, and their results show the association between endothelial cells and cardiomyocytes in human heart tissues [93,94,95]. These findings prove the established connection between cardiomyocytes and networks of endothelial cells and fibroblasts [93].

Although we are still far from a safe and functional 3D bioprinted cardiovascular construct, this field of research has been developing and improving rapidly. Future studies should focus on tackling the critical issues of cardiomyocyte maturation, improved microvasculature constructs and on the creation of a safe microenvironment for cell culture and biocompatibility. Another promising area of using 3D bioprinting technique is to design cardiac constructs for drug response and cardiotoxicity tests which could contribute to new drug development [96,97].

## 6. Three-Dimensional Bioprinting of Vascular Constructs and Grafts

Vascular grafts are vital in the surgical treatment of coronary artery disease and peripheral vascular disease, arterio-venous fistulae in haemodialysis, and repair of large-vessel aneurysms and congenital defects [98]. Autologous grafts remain the gold standard; however, 30% of patients lack suitable grafts [99,100]. Synthetic grafts suffer significantly lower patency than autologous equivalents [101] and are limited in <5 mm-vessel repairs by a high incidence of thrombosis [100]. Three-dimensional bioprinting to create tissue-engineered vascular grafts (TEVGs) offers a promising future alternative for the production of individualised grafts.

### 6.1. Requirements of a TEVG

TEVGs must be able to generate functional endothelium, be non-immunogenic, and possess similar mechanical properties to human vessels [102]. Vascular grafts should be tested for several mechanical properties. Burst pressure is the maximum pressure a vessel can withstand and is measured with burst pressure or burst probe testing [103]. Vessels must be compliant in order to accommodate pulsatile blood flow. The suture-ability of grafts is measured with suture retention testing, the maximum force required to cause the suture or graft wall to fail [104]. TEVGs also require an intact layer of endothelial cells (ECs), tested using cell viability studies [105].

### 6.2. Methods of the 3D Bioprinting of TEVGs

Biological grafts require complex structures, including branched and cellular vessels. Inkjet-based printing is either ‘vertical’ or ‘horizontal’: if the nozzle’s primary direction of movement is parallel to the vessel’s circumference, it is vertical; in horizontal printing, the primary direction of movement is parallel to the vessel’s longitudinal axis [106]. Xu et al. successfully combined vertical and horizontal inkjet bioprinting to create vessels with multi-planar bifurcations. Fibroblast cell viability was >90% after 24 h incubation. Cellular tubes (‘zigzag’ tubes) have also been successfully produced using inkjet techniques, with a cell viability of >82% after a 72 h incubation [107].

Despite these advances in inkjet bioprinting, its use for TEVGs has been limited. Inkjet printer head orifices can become clogged by bioinks of higher viscosity and the use of lower viscosity bioinks is unfavourable for mechanical strength [108], a quality essential for vascular grafts.

Hinton et al. [109] used extrusion-based bioprinting with a ‘freeform reversible embedding of suspended hydrogels’ (FRESH) support bath technique to create part of the right coronary artery, with accurate anatomical and mechanical fidelity to the 3D model. The support bath behaved as a rigid body at low shear stresses, but as a fluid at higher shear stresses, allowing for low resistance to the printer nozzle and high resistance to movement of the printed construct (Figure 4) [109].

Co-axial extrusion bioprinting utilises multiple distinct nozzles arranged coaxially, in order to print several concentric layers of biomaterial [110]. Hong et al. [111] have bioprinted vascular-like structures using a rapid-gelling gelatin–PEG–tyramine polymer with a dual coaxial nozzle. The viability of printed cells (fibroblast and endothelial cells) was 80–90%, and patency was confirmed with flushing of trypan-blue solution through the tube.

Composite bioinks can improve stability [112]; alginate and carboxymethylcellulose were combined by Milojević et al. [113] to print vessels with good biocompatibility and sufficient mechanical stability (Figure 5). Wang et al. [114] and Liu et al. [115] (Figure 6) added gelatin methacryloyl (GelMA) to alginate, producing vessels with enhanced EC proliferation, adhesion, and spreading, while the alginate sheath provided mechanical strength. Jia et al. [116] used a novel bioink of GelMA, sodium alginate, and 4 arm poly (ethylene glycol)-tetra-acrylate. The combination of three bioinks stabilised the vessel and supported the maturation of vascular cells, their triaxial printing technique was able to achieve a wide range of diameters and wall thicknesses, and the printed vessels maintained high fidelity to their models, even when printing tortuous structures.

Mandrel-based printing uses a rotating rod as the support structure for the surrounding printed vessel. Gao et al. [117] used coaxial nozzles to print hollow alginate filaments in a spiral configuration along the rotating support rod, creating a macro-channel (equivalent to the vessel lumen) and microchannels within the vessel wall. This approach produced biocompatible blood vessels with relatively strong mechanical strength. In a recent study, Jin et al. [118] used a novel approach in which nanofiber electrospinning was used to create an inner EC tube, and mandrel-based extrusion printing to create an outer layer of smooth muscle (SM) cells (Figure 7). This process yielded >90% cell viability, tensile strain sufficient to withstand blood pulsation, and suture retention strength greater than the common carotid artery. The combination of biocompatibility and mechanical strength render these processes promising for clinical application.

Xu et al. [119] created a vascular construct with both strong biocompatibility and mechanical strength using a novel integrated tissue–organ printer [120]. Two syringes of bioink were used, one for the inner layer of ECs and the other for the outer SM cells. The porosity of the GelMA bioink was varied, creating different-sized pores for each layer, essential for the different sizes of endothelial and SM cells. Gold et al. [121] disposed with a supporting sacrificial structure and used direct extrusion printing to fabricate vessels with a novel bioink: nanoengineered extracellular matrix (nECM). Their nECM comprised GelMA, poly (ethylene glycol) diacrylate, and two dimensions nanosilicates. This nECM had high printability and fidelity, cell viability >80%, and mimicked the thromboinflammatory response of native vessels.

Stereolithographic bioprinting uses patterned UV light to photocrosslink multiple layers of bioink at once. In contrast with extrusion or inkjet printing, it does not print in points; as such, it more easily produces larger-scale vascular structures [122]. Krishnamoorthy et al. [123] used this layer-by-layer approach to print vascular constructs using fibroblasts encapsulated within GelMA bioink. Cell viability after 48 h incubation was 80%. They identified an ideal cure depth (photocrosslinking thickness of an individual layer) range of 200 µm for this printing process, which gave the best shape fidelity and resolution. Thomas et al. [124] trialled multiple photoink formulations to find the ideal sacrificial photoink in which to embed ECs. A percentage of 1.5% methacrylated hyaluronic acid (HAMA) was found to be superior due to its rapid digestion time, resulting in a faster release of ECs. The authors created a sophisticated bifurcating vessel with sufficient EC proliferation (Figure 8).

### 6.3. Summary

Three-dimensional bioprinting represents an important potential solution to the shortage of biocompatible vascular grafts. Multiple techniques have been trialled in vitro, with extrusion-based printing being the most common approach. These studies demonstrate excellent viability and proliferation of printed ECs. Relatively few studies have tested the mechanical properties of printed structures which need to be robust under arterial pressure and inform future in vivo trials [113,117,118,119].

## 7. Three-Dimensional Bioprinting of Heart Valves

Valvular heart disease is a serious global health problem, affecting 3.1% of the adult population and projected to claim 23.6 million lives by 2030 [125,126]. Of the four valves in the heart, the aortic valve bears the greatest haemodynamic load, pumping 3–5 L per minute and undergoing 30–40 million cycles per year [127]. Degenerative aortic valve disease is the most common cause of valvular heart disease representing at least 65.2% of cases (aortic stenosis 47.2%, aortic regurgitation 18.0%) [125,128]. The current clinical solution for a diseased valve is replacement with a prosthetic one which can be classified as either mechanical (made from metal or carbon) or bioprosthetic (made from chemically fixed animal cardiac tissue). These prosthetic valves have transformed the treatment of valvular heart disease; however, they are not without their problems. Although mechanical valves are the most durable they require lifelong anticoagulation therapy [129,130]. Conversely, bioprosthetic valves do not require anticoagulation therapy but suffer from limited durability [130,131,132,133]. Furthermore, both types of prosthetic valves require replacement (mechanical valves ~20 years, bioprosthetic valves as short as 5 years in younger patients) which is of growing concern as the population ages, and as younger patients are being treated with the adoption of lower-risk procedures such as transcatheter valve implantation [126,134].

Tissue engineered heart valves (TEHVs) present a possible solution to this problem, with the potential to provide a valve that does not require anticoagulation therapy and could last a lifetime [135,136,137]. However, a successful TEHV has complex requirements including the ability to encourage cells to infiltrate, differentiate and proliferate in a highly specific manner, as well as exhibit adequate mechanical and haemodynamic functionality [138,139,140]. These, along with complex regulatory approvals and patient-to-patient variability means that TEHVs are yet to reach the clinic.

Bioprinting is one possible avenue to creating a successful TEHV, with the ability to construct patient-specific, hierarchical, cell-laden constructs that can also match the complex microstructure of a native valve. A handful of attempts have been made using extrusion-based, light-based and bioplotting bioprinting techniques and are described below.

### 7.1. Extrusion Based TEHVs

Extrusion-based printing can rapidly fabricate anatomically correct heterogenous TEHVs using two types of cell-free poly-ethylene glycol-diacrylate (PEGDA) which are able to support porcine aortic valve interstitial cells (PAVICs) after 21 days [141]. Furthermore, the same researchers incorporated SM cells and valve interstitial cells (VICs) into anatomically correct extrusion bioprinted valve roots and leaflets made from alginate and gelatin (Figure 9) [142]. Both SMCs and VICs were viable (>80%) after 7 days and expressed elevated alpha-smooth muscle actin (aSMA) and vimentin, respectively. Cell-laden constructs initially exhibited weaker ultimate tensile strength, failure strain and Young’s modulus than their cell-free counterparts, although over 7 days the cell-laden valves maintained their mechanical strength as opposed to cell-free constructs which weakened. Subsequently, human aortic valvular interstitial cells (HAVIC) were extrusion bioprinted into a simplified heart valve geometry (Figure 10) [143]. Different ratios of methacrylated hyaluronic acid (Me-HA) and methacrylated gelatin (Me-Gel) impacted cell spreading, mechanical properties, and printability of constructs. After three days in a static culture tube, the encapsulated cells below the surface started to remodel the hydrogel by depositing collagen and glyosaminoglycans (GAGs).

These initial studies demonstrated the feasibility of extrusion bioprinting for the creation of TEHVs, however, they did not address concerns regarding the extreme mechanical requirements. Accordingly, there have been multiple investigations to reinforce the mechanical integrity of cell-laden hydrogels. These include the incorporation of nanoparticles such as nanocrystalline cellulose [144], variations in photocrosslinking methods [145], and reinforcement with electrospun polyacrylonitrile (PAN) fibres [146]. Despite demonstrating the ability to tailor mechanical properties, none of these studies have shown mechanical properties sufficient to match those of native heart valves.

### 7.2. Light-Based TEHVs

Stereolithography printing was the first technique to successfully 3D print a TEHV, albeit cell-free [147]. However, since then there has only been one attempt at light-based bioprinting which used PEGDA:GelMA seeded with cardiac fibroblasts [148]. The valve showed 80% cell viability after 7 days, although it was not tested for haemodynamic or mechanical properties. Since then, there have been no further attempts at light-based bioprinted TEHV, perhaps due to the lack of material heterogeneity achievable with the method, or more likely due to the relatively low mechanical strength.

### 7.3. Bioplotted TEHVs

More recently, TEHVs have been fabricated using a technique that derives from extrusion-based bioprinting, called bioplotting. Here, the bioink is deposited into a medium of matching density, enabling the suspension of the bioink in 3D space. A novel bioplotting method called freeform reversible embedding of suspended hydrogels (FRESH) has been used to fabricate an entire neo-natal scale human heart exhibiting multiple anatomical structures, such as the ventricular chambers and aortic valves (Figure 11) [149]. In the same study, tri-leaflet valves were fabricated using a cell-free collagen-based ink. While this valve did not, strictly speaking, contain cells, the same technique has been used to print cell-laden vessels with diameters of 8–50 µm. It must be noted, however, that prior to mechanical testing the valves were decellularized to enhance their mechanical properties. The valve was tested in a physiological flow loop setting as per ISO 5840 and demonstrated an adequate total regurgitation fraction, although an insufficiently high effective orifice area and transvalvular pressure gradient for the aortic setting.

The same technique has been used to evaluate the recellularization potential of simplified cell-laden TEHVs in a subcutaneous rat model (Figure 12) [150]. Rat mesenchymal stem cells (rMSCs) were printed using a bioink of highly concentrated Type I collagen hydrogels (Lifeink 200) and implanted for 2, 4, 8, and 12 weeks to assess cell infiltration, inflammation, and uni-axial tensile mechanical properties. The scaffolds expressed aSMA and vimentin biomarkers, demonstrating an acceptable inflammatory response, although there was limited cellular infiltration into the scaffold. Tensile mechanical properties varied throughout the duration of the 12 weeks, potentially showing variations of scaffold resorption and remodelling, although they were statistically insignificant. Following this, a similar study was conducted accompanied by finite element computational analysis in an attempt to characterise the remodelling of mechanical properties [151]. The study concluded that the subcutaneous culture was insufficient for the replication of native tissue mechanical properties and that a dynamic cellularization environment such as a bioreactor would be necessary.

Thus, despite the promise shown in the printing of heterogenous cell-laden valves with patient-specific geometries, bioprinting of purely bioink-based TEHVs has insufficient mechanical properties. Novel methods of mechanical reinforcement such as incorporation of nanoparticles [152,153], new methods of physical or chemical crosslinking [154], and soft network composites may help [155,156].

## 8. Three-Dimensional Bioprinting of Myocardium and Heart

Injury to the heart in the form of myocardial infarction (MI) or heart failure, can be treated with medication, angioplasty and stenting, bypass surgery and lifestyle changes, however, the only remaining option for end stage disease is a heart transplant. With the well-known drawbacks that transplants entail, including donor shortage and procedure-associated risks, tissue engineering and regenerative medicine (TE&RM) approaches have been gaining ground as potential solutions.

Current views in TE&RM support the notion that the best way to achieve adequately engineered tissue capabilities is by attempting to replicate native composition, structure and properties as closely as possible [56,157]. Bioprinting technologies, with their capability to pattern cells and biomaterials with great spatial resolution, are standing out as unparalleled tools for myocardial TE&RM and will be examined in the following paragraphs and are summarised in Table 2.

Many myocardial bioprinting studies focus on the optimisation of printing conditions, bioink formulation and/or functionalisation to incorporate additional benefits. Hence, one research group formulated a bioink based on cardiac extracellular matrix (ECM) [158], while another incorporated gold nanorods (GNRs) under the hypothesis that it would beneficially contribute to the electrical conduction of the construct [159]. For their part, Erdem et al. attempted to create an oxygen-releasing bioink by adding calcium peroxide to their GelMA-based formulation, which improved survival and function of the construct under hypoxic conditions [160]. All reported good feasibility and cell viability, together with the desired effect, albeit to a limited degree.

More recently, Ahrens et al. focused on imparting cardiomyocyte (CM) alignment in their bioprinted cardiac constructs [161]. The particularity of this approach relies on the fact that the cellular component of the bioink is not dissociated cells, but rather preassembled aggregates of tens of thousands of cardiac cells, denominated anisotropic organ building blocks (OOBs), that are generated by seeding and culturing a mixture of CMs derived from iPSCs, human neonatal dermal fibroblasts and a collagen hydrogel in a micropillar array. This resulted in highly aligned, densely cellularized (+200 × 10^6^ cells/mL) cardiac struts and sheets.

The anisotropy of the bioprinted constructs came from both the auxotonic mechanical conditioning of the OOBs by the micropillars, and the shear stress direction imparted by the extrusion through the nozzle. The cardiac constructs contracted synchronously and with greater force than spheroid-based controls and could sustain pacing up to 3 Hz [161].

Similarly, Ong et al. also chose to print with cell aggregates (spheroids) rather than cell suspensions, but they did not add any supporting hydrogel under the assumption that it may hinder cell–cell interactions. Their constructs beat spontaneously and synchronously and responded to electrical pacing. When varying the ratio between CMs, ECs and cardiac fibroblasts, results suggest that the latter can delay and even block electrical conduction if present in too high a quantity [162]. When implanted in a rat MI model, the cardiac patch resulted in a smaller scar area and greater vascularization than the control (omental patch), as well as better functional output (ejection fraction and cardiac output), however these were not statistically significant [163].

Interestingly, bioprinting of cells can be complemented by the simultaneous deposition of other materials that will perform a specific function. For instance, one study incorporated a polycaprolactone (PCL) frame around the bioprinted construct to subject it to auxotonic stress during culture, as well as micro-springs that enabled measurement of the patches’ contractile force. After 4 weeks in culture, samples showed the presence of aligned and dense sarcomeres, as well as connexin 43. Moreover, the cardiac constructs displayed physiologic response when subjected to drugs known to alter contraction force and frequency, namely epinephrine and carbachol [44].

In another example, Asulin et al. combined the bioprinting of three distinct components to create a cardiac construct with integrated electronics allowing for both the sensing of the patch’s activity and its pacing. One bioink consisted of a mixture of CMs and ECM, while the other two material inks were made of polydimethylsiloxane (PDMS) and either graphite flakes for conduction, or surfactant for passivation (passivation is essential for achieving accurate spatial sensing and stimulation while reducing noise) (Figure 13) [164].

In a major step towards implant personalization, Tal Dvir et al. used patient omental biopsies to create patient-specific bioinks [165]. These consisted of a mixture of the patient’s own cells, reprogrammed into iPSCs and later differentiated into the desired phenotypes (endothelial and CMs), and the patient’s extracellular matrix, turned into a thermoresponsive collagen hydrogel. Using these they bioprinted thick (2 mm) vascularized cardiac patches, made of CMs and perfusable channels 300 µm in diameter, lined with endothelial cells. Furthermore, they sought to print constructs in a more architecturally relevant, larger size, for which they developed a support medium allowing bioink curing and construct extraction without damage to the cells. This allowed them to bioprint an anatomically accurate, miniaturized heart (20 × 18 mm) (Figure 14) [149]. The heart displayed two hollow chambers as ventricles and was surrounded by perfusable channels modelled to imitate coronary arteries.

Similarly, Noor et al. also utilized a support bath to print human CMs derived from embryonic stem cells and collagen into an ellipsoidal shell, which contracted in unison and could be paced up to 2 Hz (Figure 15) [166]. Finally, Kupfer et al. focused on bioprinting complex structures, this time adapting an MRI scan of the heart to contain an inlet and outlet channel through which unidirectional flow could be propagated. A key aspect of this work is that the authors chose to bioprint hiPSCs in situ, differentiating them from cardiac lineages after construct formation. This way, the rupture of CM-to-CM connections for bioink generation is avoided, which the authors argue results in enhanced cell density and tissue connectivity [167].

Very recently, to tackle the issues of construct size and cell survival during and after bioprinting, a six-degrees-of-freedom robotic arm was modified to serve as a bioprinter. This surpasses the capabilities of any cartesian printer, which only permits bottom-up manufacturing, and allows for printing in any direction. In this approach, tubular scaffolds in the manner of vascular conduits, through which culture medium is constantly circulated, act as a backbone onto which layers of endothelial cells and CMs are deposited. The process is iterative, consisting of rounds of bioprinting and culturing, to give time for cell attachment between the new layer and the previously deposited ones, and vascular network growth. The printing is performed in a chamber, filled either with an oil bath to ensure adhesion of the bioink to complex-shaped surfaces, or culture media. Though the authors did not demonstrate the printing of a full heart, this work provides proof of concept and shows clear potential for use in the generation of complex tissues and entire organs (Figure 16) [168].

**Table 2 biomolecules-13-01180-t002:** Summary of myocardial bioprinting approaches. Abbreviations: hiPSC—human-induced pluripotent stem cells; CM—cardiomyocytes; CB—cardiac fibroblasts; EC—endothelial cells; GelMA—gelatin methacryloyl; GNR—gold nanorods; Cx43—connexin 43; hCPCs—human cardiac progenitor cells; ECM—extracellular matrix; PCL—poly-ε-caprolactone; hESC—human embryonic stem cells; CPO—calcium peroxide; ColMA—collagen methacryloyl; SMC—smooth muscle cells; PDMS—polydimethylsiloxane.

References	Construct Form	Bioinks		Key Aspect of Study	Examined Benefits
		Cells	Hydrogel		
[162]	Patch	Spheroids of hiPSC-derived CM, CB and EC	-	Biomaterial free	Spontaneous contraction, ability to pace constructs, rudimentary vascularization, in vivo engraftment
[159]	Grid	Neonatal rat CM + CF	GelMA + alginate + GNR	GNR to improve electrical conduction	Higher Cx43 expression, higher synchronous contractile frequency than constructs without GNR
[158]	Grid, patch	hCPCs	ECM + GelMA	Cardiac–ECM specific bioink	Higher cardiac and endothelial-specific gene expression than GelMA-only constructs, retention and vascularization after in vivo implantation
[50]	Patch	Neonatal rat CM	Fibrinogen + gelatin	PCL frame to impart auxotonic mechanical stress	Cell alignment, physiologic response to drugs altering force and frequency of contraction
[149]	Ellipsoid	hESC-CM	Collagen	Ventricle-like shape	Spontaneous, synchronous contraction, pacing at 1 and 2 Hz.
[166]	Patch, two-chambered ellipsoid	hiPSC-derived CM and EC	ECM	Patient specificity, vascularization, shape	Cardiac patch with perfusable, vascular-like channels. Spontaneous and synchronous contraction
[163]	Patch	Spheroids of hiPSC-derived CM, CF and EC	-	In vivo study of patch described in (7)	Smaller scar, greater vascularization than control (omentum patch). Greater ejection fraction and cardiac output, although not significant
[160]	Grid	Neonatal rat CM + mouse fibroblasts	GelMA + CPO	Oxygen-releasing bioink	Enhanced viability and function under hypoxic conditions
[167]	Chambered ellipsoid	hiPSCs	GelMA + ColMA	Ventricular-like shape, pump-like function, differentiation after printing	Differentiation from CM, SMC and EC. Spontaneous and synchronous contraction, a physiologic response to isoproterenol, for up to 6 weeks in culture
[164]	Patch	Neonatal rat CM/hiPSC-CM	ECM, PDMS + graphite, PDMS + surfactant	Integrated electrodes for sensing and pacing	Good cell viability, spontaneous contraction and actinin expression. Sensing and pacing at 1 and 2 Hz
[161]	Struts, patch	hiPSC-CM microtissues	Fibrinogen + gelatin	High cellular density, alignment	Higher directionality, conduction velocity and force generation than spheroid-based constructs
[168]	Lining of vascular model	hESC-CM + EC	-	Ability to print in any direction	No damage in viability or activity after printing, evidence of vasculogenesis, synchronous and spontaneous contraction

## 9. Summary and Concluding Remarks

Recent developments in 3D printing technology have revolutionized medical practice by using personalised 3D printed heart and vascular models in the enhancement, diagnosis, and treatment of cardiovascular disease. Technological advancement in 3D bioprinting has great potential and is a powerful tool utilising tissue engineering principles to bioprint cardiac cells, vascular and tissue constructs. New advancements in 3D bioprinting have made it possible to construct organ-mimicking myocardial or cellular structures with high precision and flexibility. A variety of biomaterials in combination with various bioprinting methods have been shown to demonstrate great potential in 3D bioprinting cardiac constructs and grafts, myocardium and heart valves. Despite promising results and rapid progress further research and investigation is required to better understand the cross over between a native organ and amalgamation of bioprinted heterogeneous tissue constructs. Although mechanical properties are in some cases being reproduced, novel methods of mechanical reinforcement are essential to enable translation in vivo. Whilst the area of valve replacement is advancing rapidly, we see less exploration of bioprinting of vascular grafts. Whole organ bioprinting is not yet achievable due to the many challenges that exist in the inherent biology of the process and of the bioink materials and of the post-bioprinting/fabrication maturation process (comprising mechanical, electrical and perfusion) [42,169]. Bioprinting of functional tissues such as myocardium is a complex phenomenon due to their complex anatomical structures, thus in vitro fabrication of myocardial tissues requires an arrangement of multiple cell types in order, including vascular network, lymphatic vessels and muscle tissues, within engineered cardiac construct tissues [42]. As the ultimate goal in 3D bioprinting is to print the injured or damaged organ in situ, attention should be paid to the sterilization and safety of the in situ bioprinting processes [170].

Integration of nanotechnology and biopharmaceuticals along with 3D printing (nanoprinting) represents a new direction in 3D bioprinting, leading to the development of personalized nanomedicines with innovative perspectives [171,172,173]. Three-dimensionally printed microfluidic chips have made it possible to improve patient compliance by adapting to an individual patient’s anatomy through the delivery of personalized medicine. The main barriers that hinder these developments derive from the lack of clear regulatory guidelines to fabricate 3D bioprinting in clinical settings and the limited accessibility to good manufacturing practice printers [174]. As barriers and challenges are gradually overcome, 3D bioprinting will continue to play an important role in changing the treatment of cardiovascular diseases with clinical applications becoming possible in the near future.

## Figures and Tables

**Figure 1 biomolecules-13-01180-f001:**
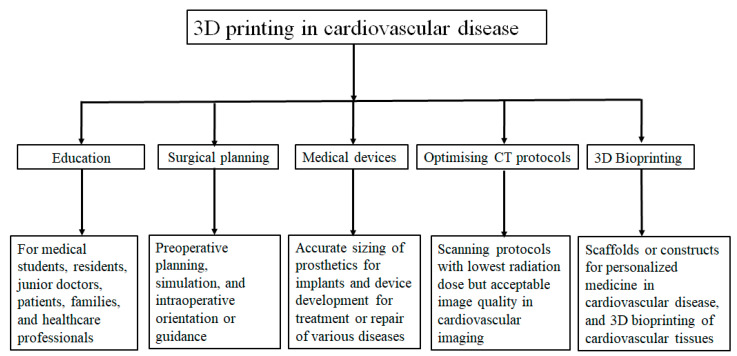
Flow chart showing applications of 3D printing, including bioprinting in cardiovascular disease. Reprinted with permission under open access from Sun [9].

**Figure 2 biomolecules-13-01180-f002:**
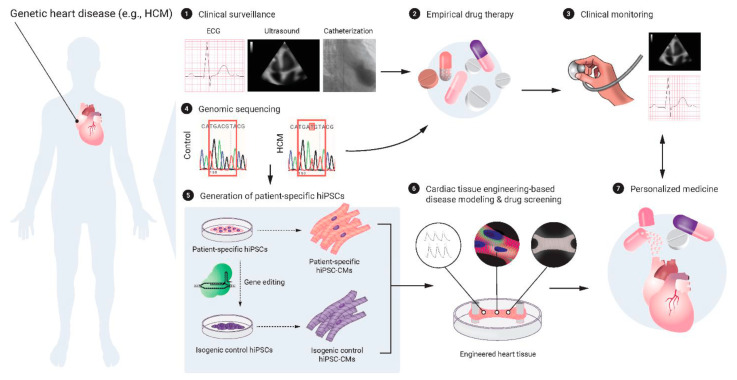
Schematic diagram highlighting the application of cardiac tissue engineering in the modelling of personalized medicine for genetic heart diseases. Reprinted with permission under open access from Häneke et al. [79]. CM—cardiomyocyte; HCM—hypertrophic cardiomyopathy; hiPSC—human inducible pluripotent stem cell.

**Figure 3 biomolecules-13-01180-f003:**
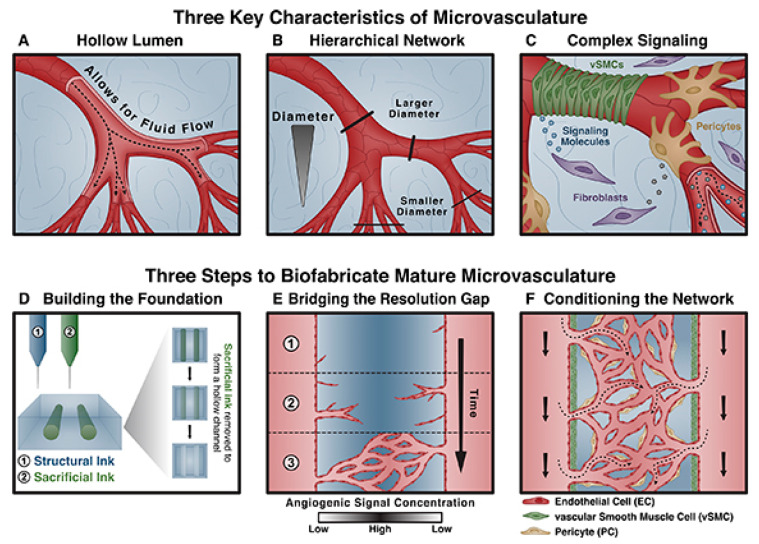
Schematic showing the key characteristics and steps to biofabricate mature microvasculature. (**A**) The presence of a hollow lumen to allow for blood flow; (**B**) hierarchy-based branched network; and (**C**) a complex signalling milieu. Seymour et al. propose three steps to biofabricate mature microvasculature involving (**D**) an extrusion-based bioprinting technique to create the hollow lumen foundation, (**E**) use of proangiogenic signalling for capillary formation, and (**F**) stabilisation of the vascular network with supporting cell types. Reprinted with permission from Seymour et al. [88].

**Figure 4 biomolecules-13-01180-f004:**
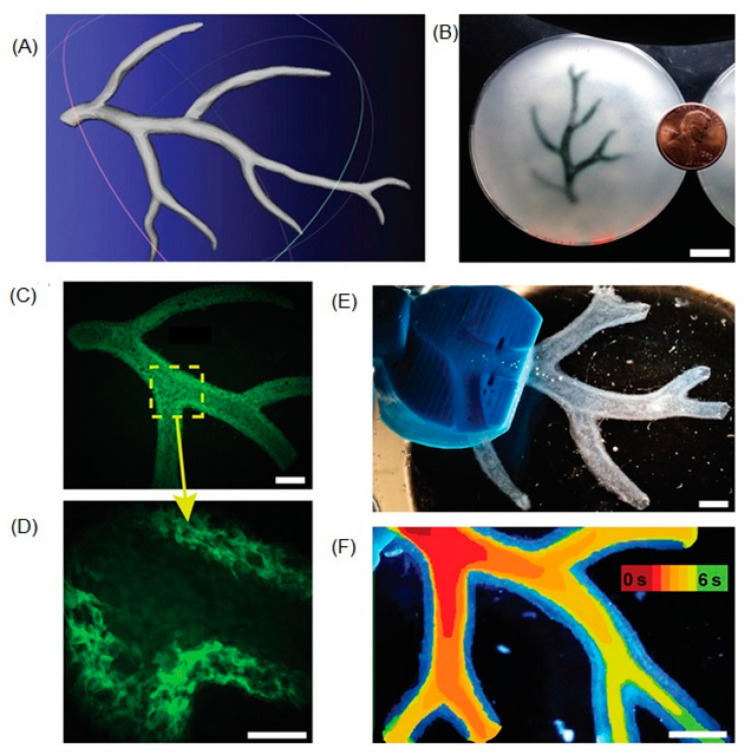
(**A**) model of a section of the right coronary arterial tree, printed with FRESH. (**B**) The arterial tree printed in alginate (black) and embedded in the support bath. (**C**) A section of the arterial trees printed in fluorescent alginate (green), demonstrating a hollow lumen. (**D**) A zoomed-in view, showing the vessel wall of <1 mm thickness. (**E**) A dark-field image of the arterial tree mounted in a perfusion fixture to position a syringe in the root of the tree. (**F**) A time-lapse image of black dye perfused through the arterial tree demonstrating the absence of leaks through the wall. Reprinted with permission under open access from Hinton et al. [109].

**Figure 5 biomolecules-13-01180-f005:**
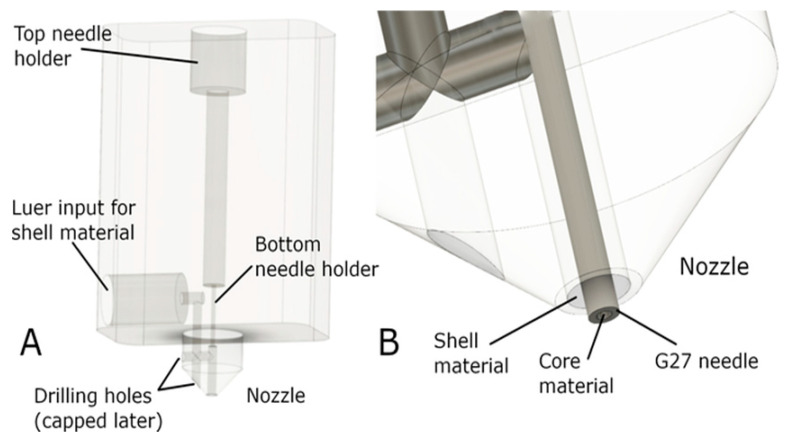
Core/shell set-up for co-axial bioprinting using composite bioinks of alginate and carboxymethylcellulose. (**A**) Overall design. (**B**) Close-up of co-axial nozzle set-up. Reprinted with permission under open access from Milojevic et al. [113].

**Figure 6 biomolecules-13-01180-f006:**
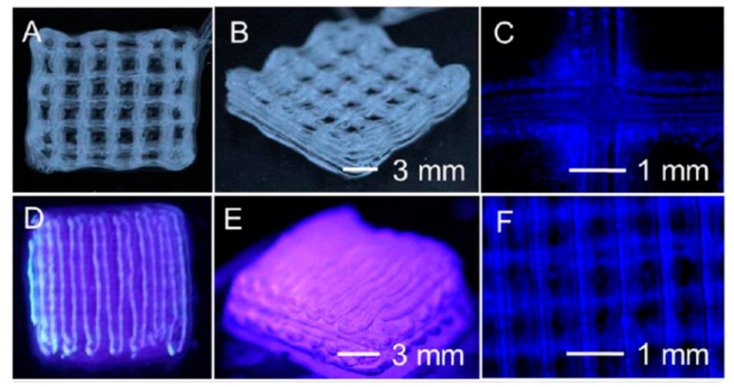
Bioprinting of 3D hollow microfibrous constructs. (**A**–**F**) A 10 layer construct with a distance between the continuous microfibers of (**A**–**C**) 2.3 mm, and (**D**–**F**) 0.0 mm. GelMA and alginate producing hollow perfusable vessels with good biocompatibility and mechanical strength. Reprinted with permission from Liu et al. [115].

**Figure 7 biomolecules-13-01180-f007:**
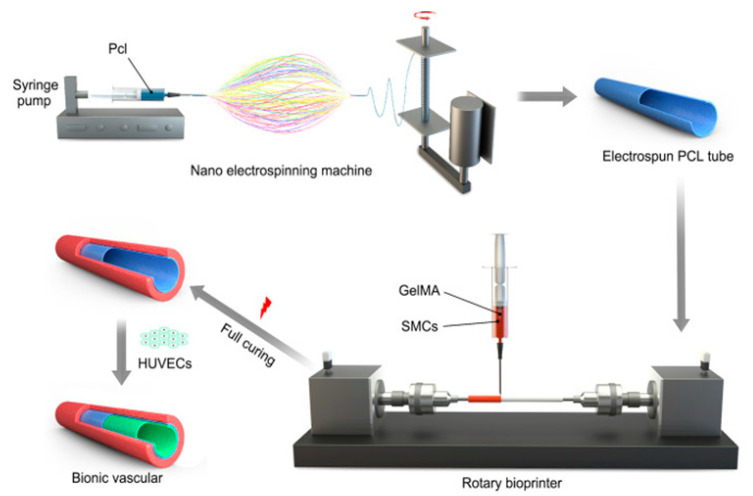
Schematic showing the novel approach for fabricating small-diameter bionic vascular tissue by combining nanofiber electrospinning and rotary bioprinting. Reprinted with permission under open access from Jin et al. [118].

**Figure 8 biomolecules-13-01180-f008:**
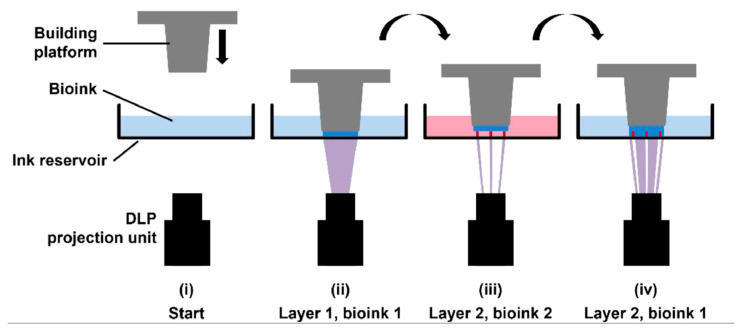
Pictorial representation of stereolithographic bioprinting with multiple inks to create the different components of the vessel wall. DLP—digital light processing. Reprinted with permission under the open access from Thomas et al. [124].

**Figure 9 biomolecules-13-01180-f009:**
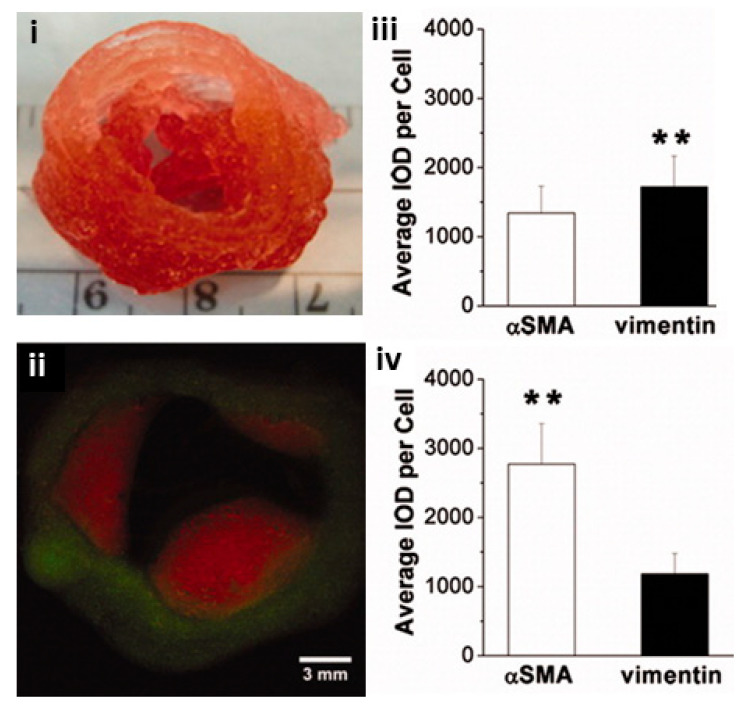
Bioprinted heart valves. (**i**) Extrusion-based bioprinted heart valve containing alginate/gelatin encapsulated with VIC/SMC, respectively, with (**ii**) fluorescent cell labelling of SMCs in green and VICs in red and the expression of aSMA and vimentin after 7 day culture and inset live/dead assay for encapsulated (**iii**) VIC and (**iv**) SMC. ** *p* < 0.01. Reprinted with permission from Duan et al. [142].

**Figure 10 biomolecules-13-01180-f010:**
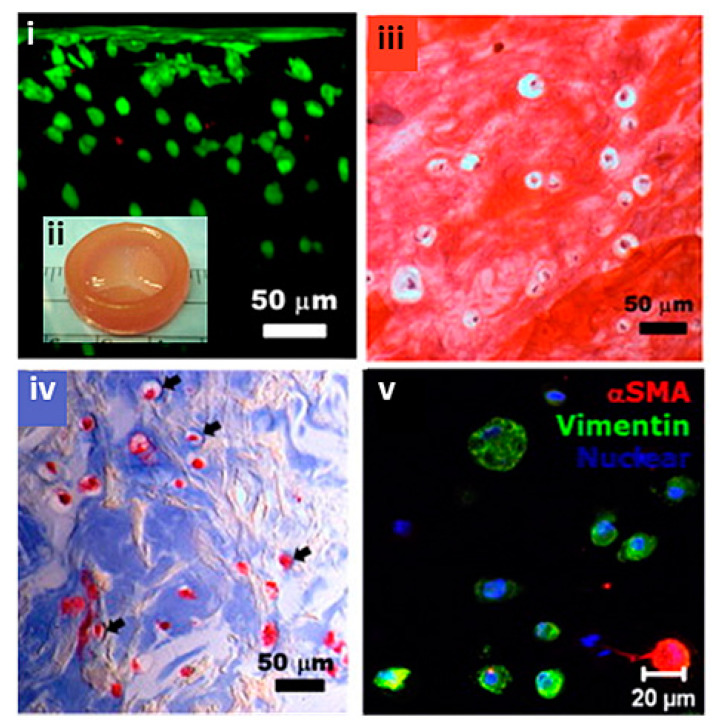
Simplified extrusion bioprinted heart valve printed from Me-Gel encapsulated with HAVIC with (**i**) live/dead staining showing cell viability through a cross section of the valve from surface to 300 um, (**ii**) inset showing simplified morphology, (**iii**–**v**) histological staining of bioprinted leaflets after 7 day culture. (**iii**) Safranin-O staining was used to stain the GAGs red, which also stained the Me-HA within the hydrogel red, (**iv**) Masson’s Trichrome staining was used to stain collagen blue. Reprinted with permission from Duan et al. [143].

**Figure 11 biomolecules-13-01180-f011:**
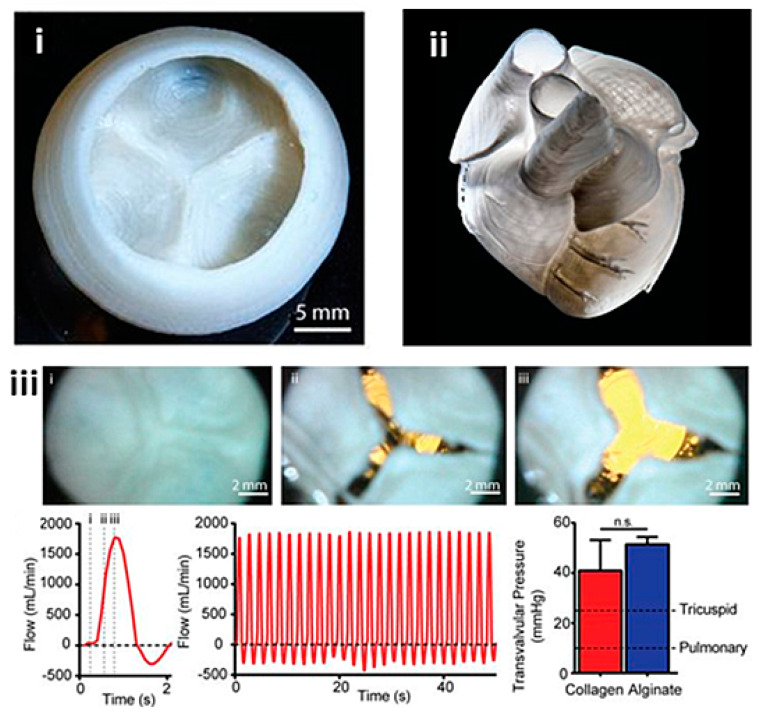
(**i**) Decellularized bioplotted TEHV from collagen-based bioink, (**ii**) entire collagen heart, (**iii**) sequence of valve opening under pulsatile flow over approximately 1 s with doppler flow velocimetry of a single cycle and multiple cycles, and maximum transvalvular pressure of printed alginate and collagen valves compared with the operating pressure for native valves. Reprinted with permission from Lee et al. [149].

**Figure 12 biomolecules-13-01180-f012:**
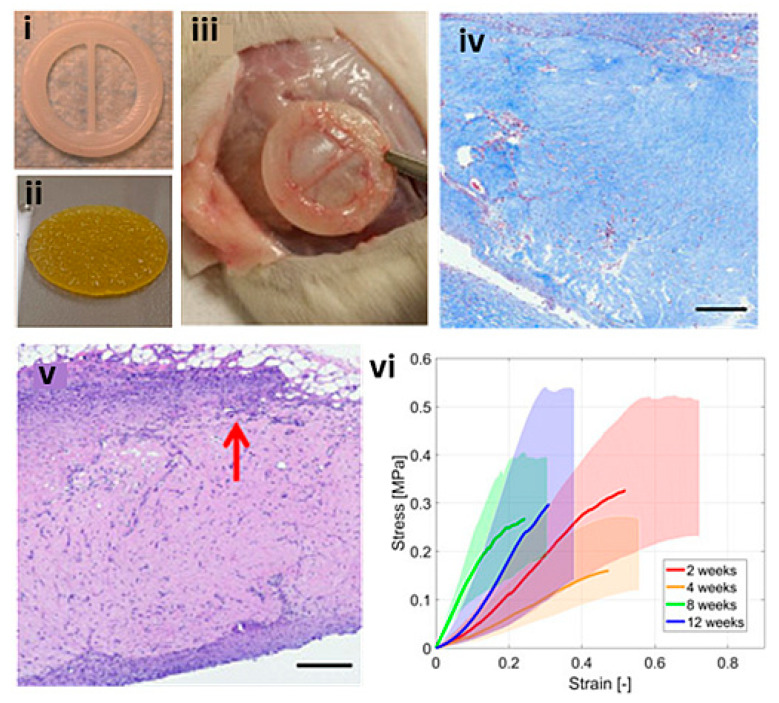
Simplified bioplotted valve made from (**i**) PCL frame and (**ii**) bioplotted type I collagen hydrogels and rMSCs, with (**iii**) showing subcutaneous explanation from Sprague–Dawley rat after 12 weeks, (**iv**) H&E staining showing increase in host cellular concentration found at the periphery (red arrow) and infiltrating within the scaffold, (**v**) Masson’s trichrome showing a diffuse blue expression representative of collagen, scale bars = 300 μm, (**vi**) stress–strain plot of the heart valve scaffolds at 2, 4, 8, and 12 weeks. Reprinted with permission under the open access from Maxson et al. [150].

**Figure 13 biomolecules-13-01180-f013:**
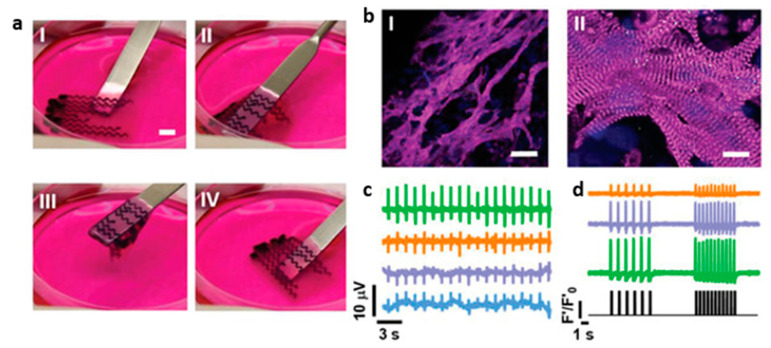
Bioprinted cardiac patch with integrated electrodes. (**a**) The patch is soft and flexible and the electrodes can be seen in black. I–IV show the patch being picked up and returned to the culture plate. (**b**) Immunostaining for sarcomeric actinin (pink) and nuclei (blue). Scale bars are 50 and 10 µm for I and II, respectively. (**c**) Recordings of action potentials from four distinct locations. (**d**) Calcium transients from three distinct locations after pacing at 7 V and 1 and 2 Hz. The pacing pattern is shown at the bottom of the panel. Reprinted with permission under open access from Asulim et al. [164].

**Figure 14 biomolecules-13-01180-f014:**
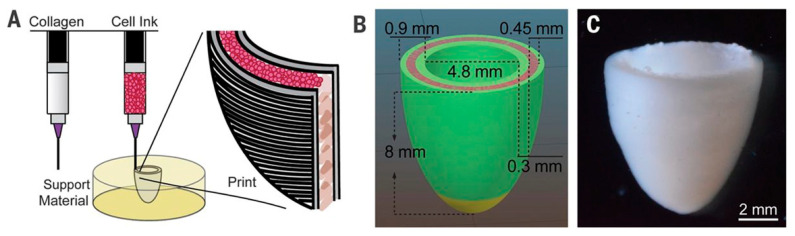
Three-dimensionally printed ventricle model. (**A**) Schematic of the dual-material printing in a support medium. (**B**) Ventricle model with the cell compartment depicted in pink, while the green represents a collagen-based bioink. (**C**) Micrograph of the bioprinted ventricle. Reprinted with permission from Lee et al. [149].

**Figure 15 biomolecules-13-01180-f015:**
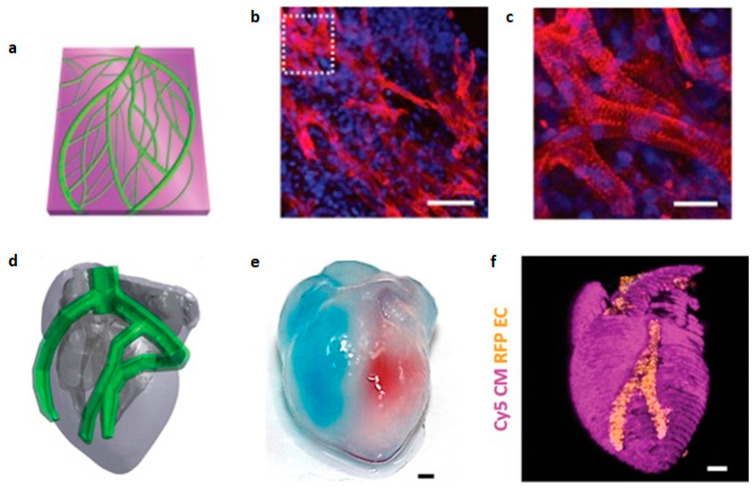
Patient-derived cardiac patch and whole heart model. (**a**) Model of the vascularised patch, the two bioinks shown in different colours. (**b**,**c**) Immunostaining for sarcomeric actinin (red) and nuclei (blue) from the patch; (**c**) is a higher magnification of the dotted area in (**b**). Scale bars correspond to 50 and 25 µm in (**b**,**c**), respectively. (**d**) Computerized model of the miniaturized heart. (**e**) After printing, the two compartments were injected with blue and red dyes to demonstrate the presence of hollow chambers and a septum in between, scale bar = 1 mm. (**f**) Confocal image of the printed heart, with CMs in pink and ECs in orange. Scale bar corresponds to 1 mm. Reprinted with permission from Noor et al. [166].

**Figure 16 biomolecules-13-01180-f016:**
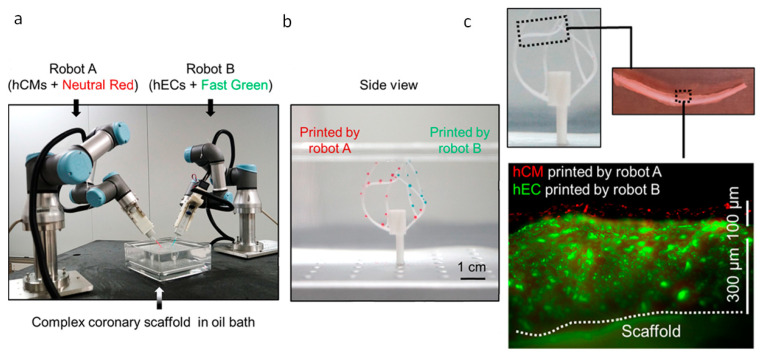
Six-degrees-of-freedom robotic arms are able to bioprint in complex shapes. (**a**) Setup with two robotic arms, one bioprinting red-stained CMs and the other green-stained ECs on top of a mould resembling the coronary tree. (**b**) All zones of the template are accessible for the robots, as demonstrated with the selectively deposited bioink droplets. (**c**) CMs and ECs are assembled in two distinct layers, bioprinted on one branch of the coronary template. Reprinted with permission under open access from Zhang et al. [168].

**Table 1 biomolecules-13-01180-t001:** Advantages and limitations of 3D bioprinting techniques. Reprinted with permission from Khanna et al. [55].

3D Bioprinting Technique	Advantages	Limitations	Ref.
IBB	Uses thermal, electromagnetic or piezoelectric technology to deposit inkjets of “ink”(materials). Rapid printing speeds and high resolution. Capable of printing low viscosity biomaterials. Availability and ease of replacement of bioinks. High cell viability and relatively low cost.	Low material viscosity (<10 Pas) and low inkjet directionality. Lack of precision with respect to inkjet size. Requirement for low viscosity bioink. Nozzle clogging and cellular distortion due to high cell density.	[74]
EBB	Ability to print biomaterials with high cell densities (higher than 1 × 10^6^ cells/mL) comparable to physiological cell densities. Can produce continuous stream of material. Can successfully print high viscosity bioinks such as polymers, claybased substrates.	Low printing resolution (> 100 μm) and slow printing speeds. Loss of cellular viability and distortion of cellular structure due to the pressure required to expel the bioink.	[75]
LBB	Rapid printing speeds and ability to print biomaterials with a wide range of viscosities (1–300 mPa/s). High degree of precision and resolution (1 cell/inkjet). Can successfully print high density of cells 10^8^/mL.	Time consuming—need to prepare reservoirs/ribbons. Lower cellular viability compared to other methods. Loss of cells due to thermal damage. SLA requires intense UV radiation for crosslinking process. Requires large amount of material. High cost. Long post processing time and fewer materials compatible with SLA.	[76,77,78]

Abbreviations: IBB—inkjet-based bioprinting; EBB—extrusion-based bioprinting; LBB—laser-based bioprinting); SLA-stereolithography.

## Data Availability

Not applicable.

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
