# Peer review of "Three-Dimensional Bioprinting in Cardiovascular Disease: Current Status and Future Directions"

_biomolecules, 2023, doi:10.3390/biom13081180_

Round 1

Reviewer 1 Report

The review article discusses the use of 3D bioprinting technologies in the field of cardiology with a focus on vascular grafts, heart valves, and cardiovascular reconstruction. First, the paper introduces cardiovascular disease and the usage of 3D printing technologies in cardiovascular research. Figure 1 summarizes these uses (i.e., surgical planning, etc.). The current progress of 3D Bioprinting in cardiology is clearly outlined in section 2 along with the associated limitations. 3D Bioprinting technologies and their advantages and disadvantages are briefly outlined in section 4. Various models for 3D bioprinting of cardiac tissues were reviewed, including hPSCs technologies for disease modeling and drug screening. Other models reviewed include myocardium, heart, vessels, and key features of the cardiovascular system. The paper delivers the objective of summarizing recent advances in 3D bioprinting and includes detailed literature review.

Suggestions for Improvement:

Figure 1 is missing an arrow that connects the first horizontal line to surgical planning and the following textbox with the words “Preoperative planning, simulation, and intraoperative orientation or guidance”.

The sentences “Although… created” in lines 183-184 should be rephrased (i.e., “inadequate maturation of tissues” instead of “immature properties of tissues created”.

Throughout the manuscript, some subjective remarks were made, which should be avoided. For example, in line 330, “with both good biocompatibility and mechanical strength” is subjective.

In line 341, “entire” should be replaced with “multiple” for clarity.

Ensure that all mentions of “in vivo” and “in vitro” are italicized (i.e., line 362).

The caption for figure 9 is very long and can benefit from being shortened.

In line 573, “in situ” should be italicized.

In table 1, “Construct shape” should be changed to “Construct form”

The last sentence of section 9, in lines 632-637 should be rephrased due to grammatical error (run-on sentence).

Quality of written English in the manuscript is adequate.

Author Response

Figure 1 is missing an arrow that connects the first horizontal line to surgical planning and the following textbox with the words “Preoperative planning, simulation, and intraoperative orientation or guidance”.

Response: Apologies. Figure 1 is replaced with a new one.

The sentences “Although… created” in lines 183-184 should be rephrased (i.e., “inadequate maturation of tissues” instead of “immature properties of tissues created”.

Response: the senence has been rephrased as suggested.

Throughout the manuscript, some subjective remarks were made, which should be avoided. For example, in line 330, “with both good biocompatibility and mechanical strength” is subjective.

Response: thank you for your comment. The manuscript is thoroughly checked with necessary changes made as suggested.

In line 341, “entire” should be replaced with “multiple” for clarity.

Response: done.

Ensure that all mentions of “in vivo” and “in vitro” are italicized (i.e., line 362).

Response: these are italicized.

The caption for figure 9 is very long and can benefit from being shortened.

Response: Figure 9’s caption has been shortened.

In line 573, “in situ” should be italicized.

Response: “in situ” is italicized.

In table 1, “Construct shape” should be changed to “Construct form”

Response: change has been made.

The last sentence of section 9, in lines 632-637 should be rephrased due to grammatical error (run-on sentence).

Reviewer 2 Report

Please add Dr Arai's research work to Kenzan method section.

Ex:Scaffold-based and scaffold-free cardiac constructs for drug testing

Kenichi Arai et all

PMID: 34233316 DOI: 10.1088/1758-5090/ac1257

Drug response analysis for scaffold-free cardiac constructs fabricated using bio-3D printer

Kenichi Arai  et all

PMID: 32487993 PMCID: PMC7265390 DOI: 10.1038/s41598-020-65681-y

My research work was tissue engineering with iPS derived cardiac myocyte using Bio3D printer. Also I focused on Vascularization like this paper. To be honest, I know most of the content of this study. This paper may be a lack of novelty. Because this is a review paper. This paper contains a lot of information which is easy to understand for us . But it may be difficult for a non specialist to understand, because this paper is unclear whether the topic is for blood vessels or heart.

This is my opinion. I believe if the author focus on the topic more preciously, the paper will get better. To do that , major revision will be needed. I don’t thind that is necessary because it’s a well written paper.

If the author don’t change the content substantially, you should ad additional research on drug analysis with Bio3D printer.

Is the paper well written? Is the text clear and easy to read? Yes, its easy to understand.

As I have mentioned above, to make the topic clear (vascular or heart ) , it is necessary to do major revision. But I don’t think we need to change it too much.

I can understand.

Author Response

Please add Dr Arai's research work to Kenzan method section.

Ex:Scaffold-based and scaffold-free cardiac constructs for drug testing

Kenichi Arai et all

PMID: 34233316 DOI: 10.1088/1758-5090/ac1257

Drug response analysis for scaffold-free cardiac constructs fabricated using bio-3D printer

Kenichi Arai  et all

PMID: 32487993 PMCID: PMC7265390 DOI: 10.1038/s41598-020-65681-y

Response: These references have been added in the citation n line 262-264.

My research work was tissue engineering with iPS derived cardiac myocyte using Bio3D printer. Also I focused on Vascularization like this paper. To be honest, I know most of the content of this study. This paper may be a lack of novelty. Because this is a review paper. This paper contains a lot of information which is easy to understand for us . But it may be difficult for a non specialist to understand, because this paper is unclear whether the topic is for blood vessels or heart.

This is my opinion. I believe if the author focus on the topic more preciously, the paper will get better. To do that , major revision will be needed. I don’t thind that is necessary because it’s a well written paper.

If the author don’t change the content substantially, you should ad additional research on drug analysis with Bio3D printer.

Is the paper well written? Is the text clear and easy to read? Yes, its easy to understand.

As I have mentioned above, to make the topic clear (vascular or heart ) , it is necessary to do major revision. But I don’t think we need to change it too much.

Response: Thank you for your comments and suggestions. As suggested, we have added the references on drug analysis using 3D bioprinting in section

Reviewer 3 Report

The manuscript offers a comprehensive review of the current state of 3D bioprinting in cardiovascular disease, with broad focuses on its applications in cardiac tissues, vascular constructs and grafts, myocardium, and heart valves. The authors detail the various techniques and strategies  and challenges associated with 3D bioprinting for cardiovascular tissues. The manuscript is well-documented and supplemented with illustrative examples.

However, there are a few areas that could benefit from further clarification or enhancement:

Introduction: The authors have broadly addressed  various cardiovascular clinical needs, explored the potential of 3D printing as a solution, described different 3D printing methods, and referred to iPSCs..... To provide the reader with a roadmap of the topics that are treated superficially or in more depth, or as an example-based review, a clear statement of the objectives in the introduction would be beneficial. The authors are encouraged to clearly state their main focus areas.

3D Bioprinting Technologies: While the authors provide a comprehensive overview of various bioprinting methods, a comparative analysis of these methods is nessesary. This analysis should underscore the advantages, disadvantages, and the suitability of each method for different applications within the field of cardiovascular disease.

3D Bioprinting Cardiac Tissues: The authors' exploration of the use of human pluripotent stem cells (hPSCs) in cardiac tissue engineering is intriguing. However, the manuscript could be enriched by introducing other cell types that have been investigated for 3D printing. Subsequently, a critical assessment of the cell type with the greatest potential for clinical translation would be a valuable addition.

Future Directions: The authors conclude by emphasizing the need for further research to better understand the relationship between native organs and bioprinted tissue constructs. However, the conclusion could be strengthened by providing more specific recommendations for future research. For instance, the authors could identify key questions that remain unanswered in the field and highlight the most promising areas for future investigation.

Author Response

The manuscript offers a comprehensive review of the current state of 3D bioprinting in cardiovascular disease, with broad focuses on its applications in cardiac tissues, vascular constructs and grafts, myocardium, and heart valves. The authors detail the various techniques and strategies  and challenges associated with 3D bioprinting for cardiovascular tissues. The manuscript is well-documented and supplemented with illustrative examples.

However, there are a few areas that could benefit from further clarification or enhancement:

Introduction: The authors have broadly addressed  various cardiovascular clinical needs, explored the potential of 3D printing as a solution, described different 3D printing methods, and referred to iPSCs..... To provide the reader with a roadmap of the topics that are treated superficially or in more depth, or as an example-based review, a clear statement of the objectives in the introduction would be beneficial. The authors are encouraged to clearly state their main focus areas.

Response: thank you for your comment. The introduction was revised to emphasise the focused areas in this review article as shown in line 92-98.

3D Bioprinting Technologies: While the authors provide a comprehensive overview of various bioprinting methods, a comparative analysis of these methods is nessesary. This analysis should underscore the advantages, disadvantages, and the suitability of each method for different applications within the field of cardiovascular disease.

Response: Table 1 is provided to provide a summary of these 3D printing techniques along with advantages and limitations as shown in line 160-165 and the new table 1.

3D Bioprinting Cardiac Tissues: The authors' exploration of the use of human pluripotent stem cells (hPSCs) in cardiac tissue engineering is intriguing. However, the manuscript could be enriched by introducing other cell types that have been investigated for 3D printing. Subsequently, a critical assessment of the cell type with the greatest potential for clinical translation would be a valuable addition.

Response: thank you for your comments. This has been revised as suggested to highlight the other cell types and the ratio of culturing different cardiac cells in the 3D printed construct, as shown in line 250-256.

Future Directions: The authors conclude by emphasizing the need for further research to better understand the relationship between native organs and bioprinted tissue constructs. However, the conclusion could be strengthened by providing more specific recommendations for future research. For instance, the authors could identify key questions that remain unanswered in the field and highlight the most promising areas for future investigation.

Response: Thank you for your suggestion. This section has been revised to emphasise the future research directions in this field, line 663-683.

Reviewer 4 Report

The authors concentrate in the article on  important topic of 3D bioprinting use in cardiovascular disease. The review presents very exact and precise overwiev of technological aspects as well current status of the art and possible future direction in this subject, from education, planning, simulation to  translation bioprinted cardiac cells, vascular and tissue constructs in clinical treatment.  The problem of possible use of 3D bioprinting   seems to be nowodays very crucial ,especially in the light of the fact that development  of 3D printing technology with cellular or biological components can be potential revolution in different fields of personalized medicine. Analysis of possible applications in cardiovascular disease treatment especially bioprinted cardiac tissues,vascular conducts, heart valves and finally myocardium and heart is very impressive. In my opinion the article should be accepted in present form however there can be more comment about possible practical use of 3D bioprinting in different cardiovascular diseases clinical treatment.

Author Response

Response: thank you for your positive feedback and comments. We agree that there are other applications of 3D bioprinting in cardiovascular diseases but our focus is on these areas that are widely studied in the literature with promising results available.